# Machine Learning Application for Medicinal Chemistry: Colchicine Case, New Structures, and Anticancer Activity Prediction

**DOI:** 10.3390/ph17020173

**Published:** 2024-01-29

**Authors:** Damian Nowak, Adam Huczyński, Rafał Adam Bachorz, Marcin Hoffmann

**Affiliations:** 1Department of Quantum Chemistry, Faculty of Chemistry, Adam Mickiewicz University in Poznan, Uniwersytetu Poznanskiego 8, 61-614 Poznan, Poland; 2Department of Medical Chemistry, Faculty of Chemistry, Adam Mickiewicz University in Poznan, Uniwersytetu Poznanskiego 8, 61-614 Poznan, Poland; adam.huczynski@amu.edu.pl; 3Institute of Medical Biology of Polish Academy of Sciences, Lodowa 106, 93-232 Lodz, Poland; rbachorz@cbm.pan.pl; 4Institute of Computing Science, Faculty of Computing, Poznań University of Technology, Piotrowo 2, 60-965 Poznań, Poland

**Keywords:** machine learning, colchicine, drug discovery, anticancer activity, QSAR, molecular docking, in silico screening

## Abstract

In the contemporary era, the exploration of machine learning (ML) has gained widespread attention and is being leveraged to augment traditional methodologies in quantitative structure–activity relationship (QSAR) investigations. The principal objective of this research was to assess the anticancer potential of colchicine-based compounds across five distinct cell lines. This research endeavor ultimately sought to construct ML models proficient in forecasting anticancer activity as quantified by the $IC_{50}$ value, while concurrently generating innovative colchicine-derived compounds. The resistance index (RI) is computed to evaluate the drug resistance exhibited by LoVo/DX cells relative to LoVo cancer cell lines. Meanwhile, the selectivity index (SI) is computed to determine the potential of a compound to demonstrate superior efficacy against tumor cells compared to its toxicity against normal cells, such as BALB/3T3. We introduce a novel ML system adept at recommending novel chemical structures predicated on known anticancer activity. Our investigation entailed the assessment of inhibitory capabilities across five cell lines, employing predictive models utilizing various algorithms, including random forest, decision tree, support vector machines, k-nearest neighbors, and multiple linear regression. The most proficient model, as determined by quality metrics, was employed to predict the anticancer activity of novel colchicine-based compounds. This methodological approach yielded the establishment of a library encompassing new colchicine-based compounds, each assigned an $IC_{50}$ value. Additionally, this study resulted in the development of a validated predictive model, capable of reasonably estimating $IC_{50}$ values based on molecular structure input.

## 1. Introduction

Machine learning (ML) methods are being investigated to speed up the discovery of new bioactive chemical structures. Current methods are aiming to propose novel chemical structures with desired properties based on already known chemical structures [1,2,3,4,5,6,7]. ML models learn from the experimentally determined biological activities and molecular descriptors or other mathematical descriptors of a chemical molecule [8]. The molecular features can be easily computed with the application of the RDKit [9] and Mordred [10] libraries, which take advantage of the linear representation of a structure, called SMILES [11]. With the gathering of all the knowledge given above, we can propose new chemical structures and assign the potential biological activity to them, which can be useful information for selection for experimental verification [12].

The objects of the studies, from the chemical point of view, are colchicine-based derivatives. These can be potentially used as anticancer treatments [13]. The biological activity feature, in this study, is defined as $IC_{50}$ (nM), which is defined as a quantitative measure that indicates how much of a particular inhibitory substance (e.g., drug) is needed to inhibit, in vitro, a given biological process or biological component by a half [14]. This study aims to predict $IC_{50}$ values for various cell lines, namely:1.A549—adenocarcinomic human alveolar basal epithelial cells, lung cancer related [15];2.BALB/3T3—detection of the carcinogenic potential of chemicals [16];3.LoVo/DX—human colon adenocarcinoma doxorubicin-resistant cell line [17];4.LoVo—human colon adenocarcinoma cell line, colorectal cancer related [18];5.MCF-7—breast-cancer-related cell line [19].

The resistance index (RI) was calculated to assess the resistance of the cancer cell lines given above. The RI index indicates how many times a resistant subline is chemoresistant relative to its non-resistant cell line. It is calculated with the application of the following formula: RI = $IC_{50}$ for LoVo/DX cell line divided by $IC_{50}$ for LoVo cell line. When the RI value is in the range from 0 to 2, the cells are sensitive to the compound, an RI in the range 2–10 means moderate sensitivity, and an RI above 10 indicates strong resistance [20].

The selectivity index (SI) serves as a metric for evaluating the selectivity of a novel colchicine-based compound. It is determined for each specific cell line through the utilization of the following formula: SI = $IC_{50}$ for the normal cell line divided by $IC_{50}$ for the corresponding cancerous cell line. An SI value greater than unity indicates that the drug exhibits enhanced efficacy against tumor cells in comparison to its toxicity towards normal cells. For instance, in the case of the MCF-7 cancer cell line, the SI is calculated as follows: SI = $IC_{50}$ for the BALB/3T3 normal cell line divided by $IC_{50}$ for the MCF-7 cancer cell line [21].

The first level of selection can be conducted in silico, as it requires a low-resource approach. Then, from the selected compounds, some can be tested in vitro. The presented cell lines can be treated with new colchicine-based derivatives and tested for the required biological activity and safety of the new compounds in vitro. The aim is to have as low an $IC_{50}$ value as possible for the cancer cell lines and higher activities for the BALB/3T3 cell line. If the colchicine-based compound has met the requirements for both good biological activity and safety, the evaluation of the compound can go to the next stage, the in vivo stage.

The data underpinning this study were gathered from diverse sources, and the compounds under scrutiny underwent rigorous in vitro assessments. Specifically, the data stem from the experimental efforts of medicinal chemists (Czerwonka, Krzywik et al.) as documented in multiple sources [22,23,24,25,26]. These sources serve as pivotal components of this study, likely containing comprehensive details about colchicine-based compounds. They encompass information regarding synthesis methodologies and the compounds’ anticancer attributes, denoted by their half-maximal inhibitory concentration ($IC_{50}$) values in nanomolars (nM) against different cell lines. Additionally, this study refers to the availability of other relevant data at the PubChem database [27], augmenting the dataset with additional insights into the compounds or their properties, thereby enhancing the research’s comprehensiveness.

The realm of machine learning (ML) spans a spectrum of methodologies, ranging from elementary approaches like linear regression [28] to more intricate models such as multiple linear regression (MLR) [29], decision trees (DTs) [30], random forests (RFs) [31], k-nearest neighbors (KNNs) [32], support vector machines (SVMs) [33], XGBoost [34], and neural networks [35,36]. The complexity of an ML model often correlates with the volume of required data. While simpler models can suffice for smaller datasets, their scope might be limited. Models exhibiting behaviors of overfitting and underfitting require careful consideration; overfitting occurs when a model learns patterns too precisely from training data but performs poorly on new data, while underfitting reflects inadequate learning to generalize on the dataset [37].

Additionally, molecular docking emerges as a crucial selection procedure for constructing new libraries of colchicine-based derivatives. The AutoDock Vina (AD Vina) algorithm [38], used in this study, boasts accuracy akin to its predecessor, AutoDock 4 (AD4) [39]. Plewczynski et al.’s studies [40] affirm that AD4 accurately redocked 93% of ligand–protein complexes. Despite the relatively simple scoring functions, software of this nature proves invaluable as a supporting tool in drug design processes.

Our study can be divided into two pathways, which can be run separately. The first route is the construction of new colchicine-based compounds with the application of a previously prepared algorithm [4,12]. This pathway lets us create a library of new colchicine-based structures after a few steps of selection. The second pathway is related to the biological activity, using machine learning (ML) models’ predictions. To obtain this target, we need to build and test several ML models and select the “best” of them for each of the biological activity targets. Then, with the “best” models, we can assign target features, $IC_{50}$ values, for the newly created colchicine-based structures. At the end, we have a library of in silico-tested compounds that have a predicted $IC_{50}$ value, binding affinity to the 1SA0 protein domain [41], and SYBA score [42], which indicates the potential difficulty of synthesis. The selection of the 1SA0 protein domain is predicated on the recognition that colchicine exhibits binding affinity to β-tubulin, consequently inducing microtubule destabilization by rendering the colchicine-bound dimers incapable of assembly. Notably, in eukaryotic cells, which are characterized by the expression of various isotypes of β-tubulin, βI (one of the isoforms of β-tubulin) consistently stands out as the predominant and most prominent target for drug binding interactions [20].

## 2. Results and Discussion

### 2.1. Training Data

The training dataset methodology proposed can be efficiently used for small datasets. This lets us use machine learning techniques for issues that are new, and thus, for which there is a lack of data, meaning chemical structures with assigned biological activity. This simple workaround gives us the opportunity to create new chemical structures from a small starting database [12]. Appendix A are related to this section.

### 2.2. Generative Neural Network

The generative RNN method [12] is illustrated conceptually in Figure 1. This technique enables the creation of a diverse library of novel colchicine-based structures by leveraging existing ones. By employing this approach, we have been able to develop a repository of fresh colchicine-based structures, which can undergo additional processing. For instance, selection procedures can be applied to refine and enhance these structures further.

With the 120 starting structures, we trained the neural network, namely, the recurrent neural network (RNN), with the goal of chemical space exploration. This let us teach the RNN how to reconstruct chemical structures, and with its application, new structures were proposed (Appendix A). Figure 2 shows the decreasing loss value, indicating that the model’s performance becomes better and better as the learning time increases (Appendix A).

The proposed approach enabled us to create 1786 chemical structures (Appendix A). Their creation was quite random, and due to that, data selection (Section 2.4 and Section 3.4) was performed. These structures are distinct from each other. This let us keep more similar structures to the starting ones, as machine learning model predictions are more accurate in the closer chemical space.

Selected structures are shown below (Table 1). The table indicates the capabilities of a generative machine learning model, the recurrent neural network (RNN) [4]. Table 1 contains two starting structures with experimentally measured $IC_{50}$ values for various cell lines and their SMILES codes. The second part of Table 1 shows some of the newly created chemical structures, their Tanimoto similarity [43] to the starting structure above, and, following the application of trained machine learning models, their predicted $IC_{50}$ values. It can be observed that a small change in structure may lead to profound changes in biological activity (Appendix A).

### 2.3. Machine Learning Models for Anticancer Activity

The biological activity, given by $IC_{50}$ (nM), was transformed with Equation (Equation 1) (Section 3.3). This is shown in Appendix A, and the results were saved in Appendix A. The correlation of the molecular descriptors [46] with the $IC_{50}$ values is shown in Appendix A.

Five different machine learning methodologies were applied (Appendix A). Namely, multiple linear regression (MLR) [29], decision tree (DT) [30], random forest (RF) [31], k-nearest neighbors regression (KNN) [32], and support vector machines (SVMs) [33]. Each of them was investigated through various molecular descriptors as training features. The lower the number of features, the higher the correlation of the features to the biological activity parameter, given by $IC_{50}$ (nM).

Figure 3, Figure 4, Figure 5, Figure 6 and Figure 7 (created within Appendix A) exhibit the quality measurements for each methodology investigated across individual cell lines. The term ‘random state’ [47] denotes the initial seed employed by a pseudo-random number generator (PRNG) within machine learning algorithms. Its function is to ensure reproducibility by establishing a consistent starting point each time the code is executed, thereby ensuring uniform results during experiments or model training.

The random state value signifies variations in both the training and testing data while maintaining their specified percentages, as detailed in Section 3.3. The distribution of these datasets can be observed in Appendix A. The utilization of different random states during data splitting leads to alterations in the composition of the training and testing datasets. Consequently, distinct data points are employed for the training and testing procedures at each random state.

Two scoring parameters were explored to select the best final models for predicting the activity of each of the cell lines. The first parameter is the correlation coefficient (R) [48], and the second is the root mean square error (RMSE) [49]. According to a study conducted by D. Chicco et al., the coefficient of determination $R^{2}$ (where the correlation coefficient is the square root of $R^{2}$) provides more informative insights than the RMSE. $R^{2}$ does not possess the interpretative limitations of the MSE, RMSE, MAE, and MAPE. Therefore, we suggest using R-squared as the standard metric for evaluating regression analyses in any scientific domain [50]. Conversely, B. Lucic has demonstrated that R can present an overly optimistic measure of agreement between predicted values and corresponding experimental values, leading to an excessively optimistic conclusion about the model’s predictive ability [51]. The RMSE provides insights into absolute prediction errors, which could be more informative in practical scenarios compared to R-squared, which focuses on explaining variance. Since our aim is to select the best machine learning model for each cell line, both parameters were taken into consideration.

As Figure 3, Figure 4, Figure 5, Figure 6 and Figure 7 depict, we can draw the following conclusion: The “best” methodology for determining each of the cell lines’ biological activity is the random forest (RF) [31] methodology. In every instance, it exhibits the highest correlation coefficient across both the training and testing datasets, along with the lowest RMSE. This explains why the RF methodology was chosen as the final model for each of the biological activities, as the RF methodology learns best how to generalize on the given dataset. The “best” models are shown in Appendix A, respectively, for each of the cell lines.

Bias in ML algorithms skews results in favor of or against an idea, in either direction. This is a systematic error brought on by false assumptions made throughout the ML learning process. This can have an impact on how an ML model is built [52].

The biological activity, given in $IC_{50}$ (nM), for the A549 cell line machine learning investigation results are stored in Appendix A. Figure 3 depicts the “best” performance of simple machine learning models at the lowest number of features (five features). The two best methodologies observed are DT and RF. It should be stated that the RF model is composed of many DT models. The worst-performing approaches are the other ML methods employed (MLR, KNN, SVM).

The R values (higher values indicate better performance) of the DT and RF ML models on the training datasets are similar, but RF demonstrates superior performance on the test set of data. While other ML models perform adequately on the training data, they significantly underperform on the testing data, as evident from their R values, indicating underfitting [37] to the test data. The RMSE values, depicted in Figure 3, suggest that DT and RF are the most effective methods for predicting the A549 $IC_{50}$ (nM) parameter. These ML models exhibit the lowest RMSE values compared to MLR, KNN, and SVM. Both the R values and RMSE consistently support the use of a random state of 15 in the RF model as the optimal choice. Details of the data splitting for this model can be found in Appendix A.

The molecular descriptors utilized for predicting A549 $IC_{50}$ (nM) values include the following: AMID_O (average molecular ID on O atoms), EState_VSA5 (EState VSA Descriptor 5 with a range of 1.17 to less than 1.54), MDEO-12 (molecular distance edge between primary O and secondary O), SaasC (sum of aasC), and VSA_EState5 (VSA EState Descriptor 5 (5.74 ≤ x < 6.00)) [53]. The rationale behind their selection is detailed in Appendix A.

**Figure 3 pharmaceuticals-17-00173-f003:**
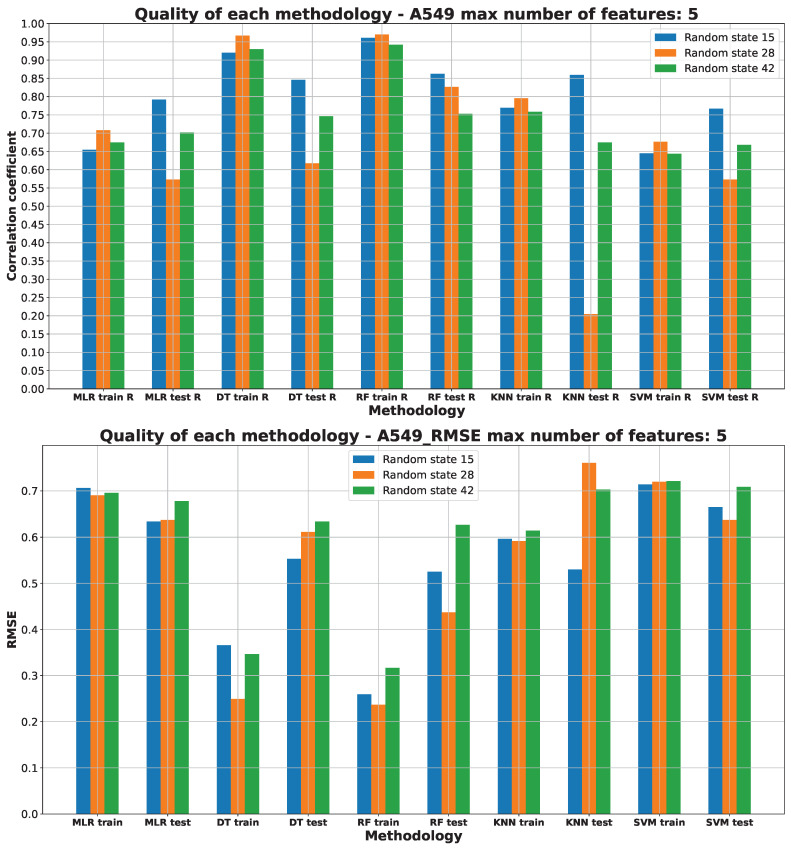
Machine learning models’ performance on A549 cell line.

The biological activity, given in $IC_{50}$ (nM), for the BALB/3T3 cell line machine learning investigation results are stored in Appendix A. Figure 4 depicts the “best” performance of simple machine learning models at the lowest number of features (six features). The two best methodologies are observed to be DT and RF. The worst-performing approaches are the MLR, KNN, and SVM machine learning approaches.

The R values for the DT and RF ML models on the training datasets show a comparable performance, with RF slightly outperforming DT on the test set. Other ML models perform reasonably well on the data, but are worse than RF model. The KNN is reflecting the signs of underfitting [37] to the testing data.

The RMSE values presented in Figure 4 highlight that the most effective models for predicting the BALB/3T3 $IC_{50}$ (nM) parameter are DT and RF. These ML models show the lowest RMSE values compared to other methods such as MLR, KNN, and SVM. Both the R values and RMSE consistently endorse the RF model with a random state of 15 as the optimal choice. Detailed information regarding the data splitting for this model is available in Appendix A.

The molecular descriptors used to predict the BALB/3T3 $IC_{50}$ (nM) values are as follows: AMID_O (averaged molecular ID on O atoms), EState_VSA5 (EState VSA Descriptor 5 (1.17 ≤ x < 1.54)), GATS2c (Geary coefficient of lag 2 weighted by Gasteiger charge), MDEO-12 (molecular distance edge between primary O and secondary O), NdssC (number of dssC), and VSA_EState5 (VSA EState Descriptor 5 (5.74 ≤ x < 6.00)) [53]. The rationale behind their selection is detailed in Appendix A.

**Figure 4 pharmaceuticals-17-00173-f004:**
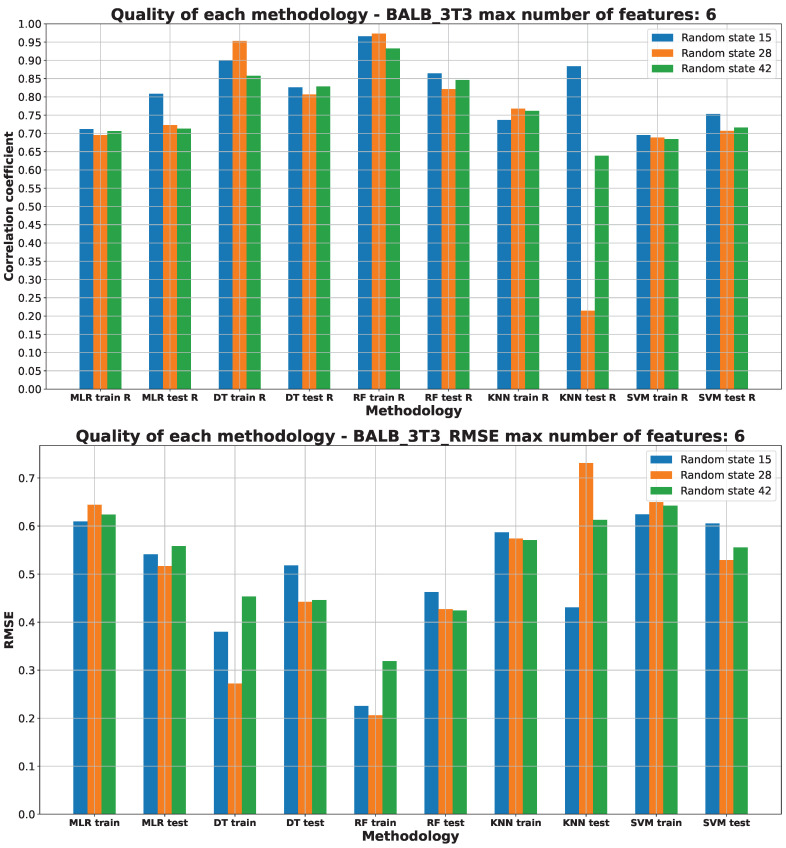
Machine learning models’ performance on BALB/3T3 cell line.

The biological activity, given in $IC_{50}$ (nM), for the LoVo/DX cell line machine learning investigation results are stored in Appendix A. Figure 5 depicts the “best” performance of simple machine learning models at the lowest number of features (five features). The two best methodologies are observed to be DT and RF. The DT methodology training correlation coefficient equals 1.00, but this methodology was not chosen as the final one due to the overfitting [37] to the training data. The final method is RF with random state 42, which performs quite well on training and testing data.

The R values of the DT and RF ML models on the training datasets exhibit a similar performance, with RF displaying slightly superior performance on average when evaluated on the test dataset. Notably, the DT models reveal signs of overfitting [37], as indicated by R values of 1.00, leading to comparatively larger errors based on RMSE compared to the RF ML model. Interestingly, the RMSE for the DT training dataset with a random state of 28 is notably low, primarily due to overfitting [37] to the training data.

**Figure 5 pharmaceuticals-17-00173-f005:**
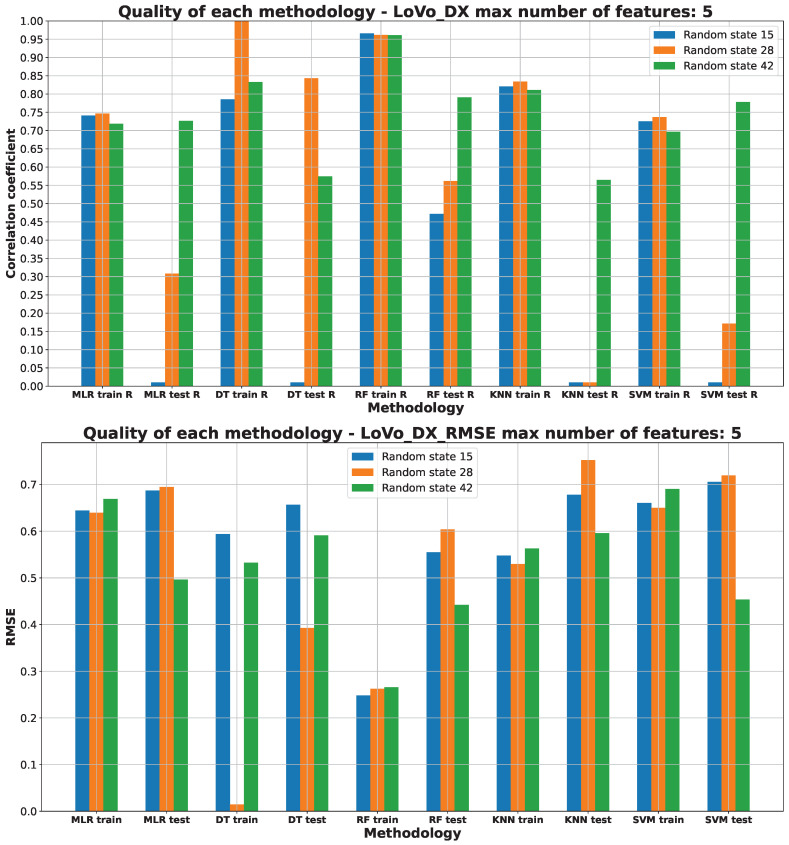
Machine learning models’ performance on LoVo/DX cell line.

While the other ML models perform reasonably well on the training data, they demonstrate notably inferior performance on the testing data, evident from their R values, signifying potential underfitting [37] to the test data. The RMSE values depicted in Figure 5 suggest that DT and RF are the most effective methods for predicting the LoVo/DX $IC_{50}$ (nM) parameter. These ML models exhibit the lowest RMSE values compared to other models like MLR, KNN, and SVM.

Both the R values and RMSE consistently support the RF model with a random state of 42 as the optimal choice. Detailed information regarding the data splitting for this model can be located in Appendix A.

The molecular descriptors used to predict LoVo/DX $IC_{50}$ (nM) values are the following: GATS2c (Geary coefficient of lag 2 weighted by Gasteiger charge), MATS2c (Moran coefficient of lag 2 weighted by Gasteiger charge), NdssC (number of dssC), RNCG (relative negative charge), and TopoPSA(NO) (topological polar surface area (use only nitrogen and oxygen)) [53]. The rationale behind their selection is detailed in Appendix A).

The biological activity, given in $IC_{50}$ (nM), for the LoVo cell line machine learning investigation results are stored in Appendix A. Figure 6 depicts the “best” performance of simple machine learning models at the lowest number of features (two features). The two best methodologies are observed to be DT and RF. As the performance of the DT and RF methodologies is similar, by our choice, the RF methodology was used to build a final predictive model, as it can cover a wider space of predictive possibilities compared to the simple DT model. The worst-performing approaches are the MLR, KNN, and SVM machine learning approaches.

The R values, indicating model performance, show comparable results between the DT and RF ML models on the training datasets, with RF demonstrating a slightly better average performance on the test dataset. Interestingly, while other ML models perform reasonably well on the training data, they notably underperform on the testing data, as indicated by lower R values, hinting at potential underfitting [37] to the test dataset.

In terms of predicting the LoVo $IC_{50}$ (nM) parameter, the RMSE values displayed in Figure 6 highlight that DT and RF emerge as the most effective methods. These ML models display the lowest RMSE values compared to alternative models such as MLR, KNN, and SVM, signifying their superior predictive accuracy.

The advantage of RMSE is also apparent in Figure 6, where the correlation coefficient alone proves insufficient as a quality parameter. The RMSE indicates that the models are comparable in terms of quality, aligning with its assessment, unlike the sole reliance on the correlation coefficient (R).

Consistently, both the R values and RMSE support the RF model with a random state of 28 as the optimal choice. For more detailed information regarding the data division used for this model, please refer to Appendix A.

The molecular descriptors utilized for predicting the LoVo $IC_{50}$ (nM) values include the following: EState_VSA5 (EState VSA Descriptor 5 (1.17 ≤ x < 1.54), and MDEO-12 (molecular distance edge between primary O and secondary O) [53]. The rationale behind their selection is detailed in Appendix A).

**Figure 6 pharmaceuticals-17-00173-f006:**
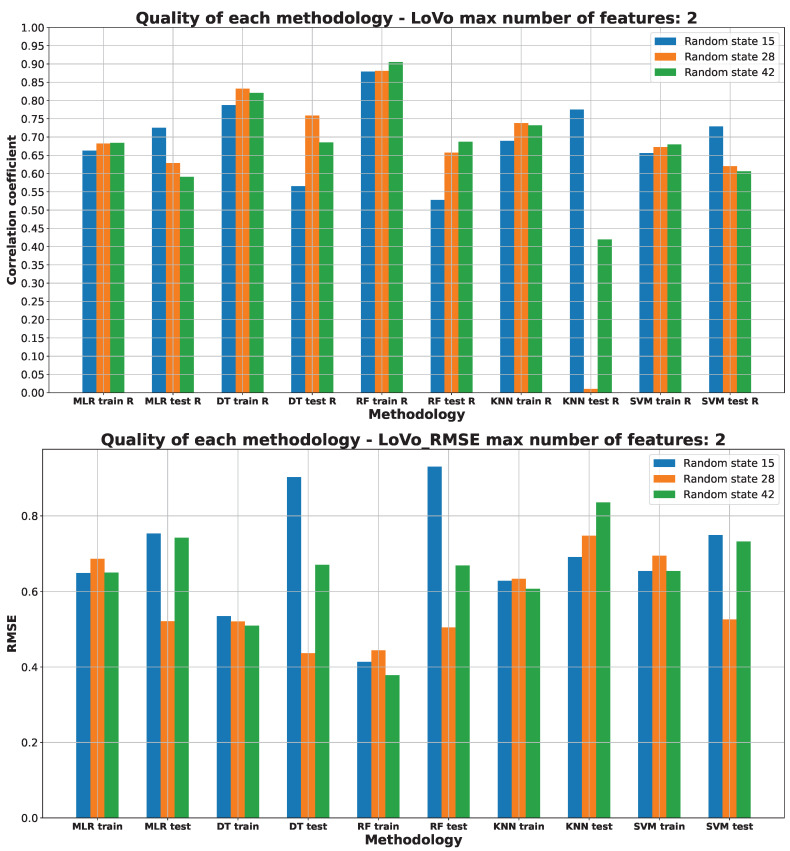
Machine learning models’ performance on LoVo cell line.

The biological activity, given in $IC_{50}$ (nM), for the MCF-7 cell line machine learning investigation results are stored in Appendix A. Figure 7 depicts the “best” performance of simple machine learning models at the lowest number of features (four features).

While the R values for the DT and RF ML models on the training datasets are fairly similar, RF demonstrates superior performance on the average test set. Notably, the performance of other ML models appears robust on the training data but markedly deteriorates on the testing data (KNN), revealing significant underfitting [37] issues, as indicated by lower R values.

Analyzing the RMSE values depicted in Figure 7, it becomes evident that DT and RF stand out as the optimal methods for predicting the LoVo $IC_{50}$ (nM) parameter. These ML models exhibit the lowest RMSE values compared to alternative models such as MLR, KNN, and SVM, highlighting their superior predictive accuracy.

The benefit of the RMSE is observable in Figure 7, highlighting the inadequacy of solely using the correlation coefficient as a quality measure. The RMSE indicates comparability among the models in terms of quality, supporting its evaluation, unlike relying solely on the correlation coefficient (R).

Consistently, both the R values and RMSE endorse the RF model with a random state of 15 as the most effective choice. For a detailed breakdown of the data splitting used for this model, please refer to Appendix A.

The molecular descriptors used to predict MCF-7 $IC_{50}$ (nM) values are the following: AMID_O (averaged molecular ID on O atoms), EState_VSA5 (EState VSA Descriptor 5 (1.17 ≤ x < 1.54), MDEO-12 (molecular distance edge between primary O and secondary O), and EState_VSA6 (EState VSA Descriptor 6 (1.54 ≤ x < 1.81)) [53]. The rationale behind their selection is detailed in Appendix A).

**Figure 7 pharmaceuticals-17-00173-f007:**
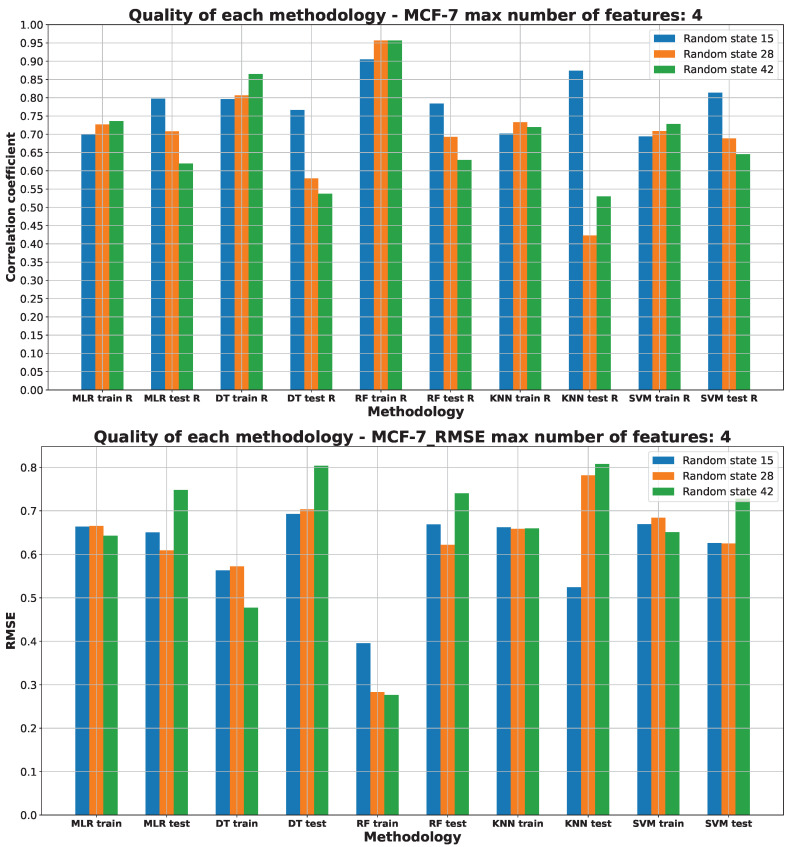
Machine learning models’ performance on MCF-7 cell line.

The final model quality measurements are presented in Table 2. The table presents the machine learning methodology selected for the final predictive model construction, the number of features used to build the model, the correlation of molecular descriptors to the target data, the overall R score [48] (calculated by asking the model to predict the target value on the whole dataset), the mean squared error (MSE) [54], the mean absolute error (MAE) [55] and, lastly, the root mean square error (RMSE) [49], all calculated on the whole dataset.

### 2.4. Data Selection

The first step of selection was conducted using Tanimoto similarity [43,56,57] (Section 3.4). This step let us cut out 430 structures that were too different in the meaning of chemical similarity calculated based on molecular fingerprints [56]. A total of 1356 new structures remained after the first step (Appendix A).

The second step of selection was the employment of the SYBA classifier [42] (Section 3.4). This provided an estimation of the potential difficulty of synthesis. Figure 8 shows the chemical space of the considered species with blue-marked starting structures and red-marked new structures. As can be seen in Figure 9, after the application of the SYBA classifier, only structures more similar to the starting ones remain. This step sieved out 1021 structures, leaving 335 structures (Appendix A).

In the final step of the selection process (Section 3.4), we employed a rigorous selection criterion for the absolute configuration of the compounds, as the starting structures have an ‘S’ absolute configuration on the seventh carbon atom [22,23,24,25,26,58]. Figure 10 provides a visual representation of the distribution of absolute configurations in the initial set of structures. Notably, all of these structures exhibited the ‘S’ absolute configuration. Subsequently, Figure 11 illustrates the distribution of absolute configurations in the newly generated colchicine-based structures. The application of this selective criterion resulted in the elimination of 50 structures from the initial dataset, reducing the total number of structures under examination to 285, as shown in Appendix A. This stringent selection process ensured that only structures conforming to the desired ‘S’ absolute configuration on the seventh carbon atom were included in our study.

The PubChem search (Appendix A) provided information that four structures out of the AI-created colchicine-based structures were found in the PubChem database of known compounds. Their (PubChem Compound Identifications) CIDs were 6351355, 162648725, 164628185, and 162672356, respectively. This indicates that the vast majority of the AI-proposed structures are new, and from them, we can pick the best candidates based on machine learning models and molecular docking. It also shows the generative capabilities of the machine learning models proposed.

### 2.5. Molecular Docking

Molecular docking studies conducted indicate that the AI-proposed structures have similar affinity to the 1SA0 (PDB ID) protein domain. Figure 12 depicts the distribution of the affinity of selected (285) structures to the protein domain. The first red line shows the raw colchicine affinity (−8.2 kcal/mol) and the second natively present structure in the raw PDB file (−8.6 kcal/mol). The combination of biological activity prediction based on ML models combined with molecular docking can enhance the selection of structures for experimental verification and potentially speed up new bioactive system discoveries. Of the structures, 131 have an affinity greater than −8.2 kcal/mol, and 78 have an affinity lower than −8.6 kcal/mol. The number of structures with an affinity equal to −8.6 kcal/mol is 14, and 15 structures have an affinity equal to −8.2 kcal/mol.

Figure 13, Figure 14, Figure 15 and Figure 16 show the interactions that can be formed between colchicine-based structures and the binding site of 1SA0. The green solid lines indicate hydrophobic contact, and the black dashed lines indicate possible hydrogen bond formation. All of them are stabilizing ligands in the pocket of the binding site of the protein. Indeed, molecular docking is a good tool to enhance the possibilities of biological activity mechanism explanation [59].

Additional 3D interaction visualizations between ligands from Figure 13, Figure 14, Figure 15 and Figure 16 can be found in the Appendix A, called additional supplementary figures. Appendix A illustrates the native 1SA0 ligand situated within the protein’s binding site. Appendix A displays AI-generated ligand number 73 occupying the binding site of the 1SA0 protein. Likewise, Appendix A exhibits AI-generated ligand 113 also within the identical binding site of the same protein. In a similar vein, Appendix A portrays AI-created ligand 162 occupying this binding site. Appendix A depict all the previously mentioned ligands simultaneously occupying the binding site within the 1SA0 protein domain. The visualizations were performed using the Chimera tool, version 1.16 [60].

Molecular dynamic simulations (MDSs) play a crucial role in advancing molecular biology and facilitating the discovery of new drugs. However, this study did not encompass this aspect, leaving ample space for undeniable enhancements to the presented method.

## 3. Materials and Methods

The files which are mentioned in the manuscript are attached in the Appendix A, and their names are stored in the “Files” attachment. The overall workflow is presented in Figure 17. The files pertaining to the conducted study have been consolidated on GitHub’s platform and are accessible through this link: https://github.com/XDamianX-coder/Colchicine_ML (accessed on 18 January 2023). All files can also be viewed within the compiled Appendix A containing all the code and Jupyter notebooks, and Appendix A, where the Excel (Version 2312) results are stored. The file numbering has been maintained to ensure the project’s readability.

### 3.1. Training Data

The training data for both the generative neural network creation and creating the machine learning models for anticancer activity prediction were collected from previously published material [22,23,24,25,26]. From this material, we collected 120 structures, SMILES codes [11], with experimentally assigned $IC_{50}$ activities towards various cell lines, such as A549, BALB/3T3, LoVo/DX, LoVo, and MCF-7. All the data can be viewed in Appendix A.

Based on this collection, we proposed new chemical structures and their estimated activity for each of the evaluated cell lines. It also should be stated that the $IC_{50}$ parameter unit in this case is given in units of nM for both the starting and newly created structures.

We have created various SMILES [11] representations of each of the colchicine derivatives. This was performed in Appendix A was created as the training dataset, with 118,070 SMILES representations.

### 3.2. Generative Neural Network

The vectorization procedure is necessary for feeding the neural network with the chemical structures’ data, as chemical structures are not an easily accessible representation for computers, but vectors are. Vectorization is the process of converting a computer-unreadable representation of data through mathematical processing into computer-readable objects known as mathematical vectors [61]. The format of the chemical representation that was vectorized was the so-called molecular sequence [4], which was derived from the SELFIES [62] depiction of a molecule. This was conducted within Appendix A.

The generative neural network was built with the application of the previously proposed architecture [4,12] with little modification. As we did not have thousands of structures, we encoded and vectorized the same molecule many times using different SMILES code. This was achieved with the support of the RDKit library [9]. In that way, we were able to create a sufficient number of data points (118,070) for neural network training (106,263) and validation (11,807). This was performed inside Appendix A. The model is stored in Appendix A. The neural network performance is recorded in Appendix A.

With the application of the neural network, we proposed a number of new structures (Appendix A), which are close in the meaning of the chemical space, as our recurrent neural network (RNN) [1,2,3,12] learns how to reconstruct vectorized chemical structures. The quality of the model and the loss function were measured using categorical cross-entropy [63]. The structures proposed by the machine learning model are stored in Appendix A.

### 3.3. Machine Learning Models for Anticancer Activity

Based on the collected data (120 data points), we proposed very simple models for revealing the biological activity of unknown compounds. The following methodologies were investigated: multiple linear regression (MLR) [29], decision tree (DT) [30], random forest (RF) [31], k-nearest neighbors regression (KNN) [32], and support vector machines (SVMs) [33].

Firstly, the target features, biological activities, were transformed with the following equation: (1)$$pIC_{50}=-1\times log\left(\frac{IC_{50}}{10^{9}}\right)$$
This was performed to reduce the skewness of the data points (see Appendix A). The transformed data are stored in Appendix A.

Then, the correlation of molecular descriptors [46], calculated with RDKit [9] and Mordred [10], to each of the biological activities was provided in the nM unit of $IC_{50}$. This was conducted in Appendix A. It gives us information about the number of features that can be used at certain thresholds of correlation between molecular descriptors and biological activity.

The machine learning methodologies were investigated with three different random seed values, namely, 15, 28, and 42, which were randomly chosen. Each of the models evaluated was tested with different correlation thresholds of biological activity, $IC_{50}$ values towards certain cell lines:1.A549 (Appendix A);2.BALB/3T3 (Appendix A);3.LoVo/DX (Appendix A);4.LoVo (Appendix A);5.MCF-7 (Appendix A).
and molecular descriptors. The higher the correlation value, the fewer molecular descriptors were available for model construction. We aimed to construct the simplest models possible for each of the target cell lines.

At the end, the “best” models were chosen based on quality measurements, such as correlation threshold, mean squared error (MSE) [54], mean absolute error (MAE) [55], and root mean square error (RMSE) [49] (Appendix A). Then, the final models were built (Appendix A) and saved for each cell line separately (Appendix A). The model was used to predict the biological activities of the newly generated structures, and the outputs were recorded (Appendix A).

### 3.4. Data Selection

As the neural network can create a variety of structures, subsequent data selection must be performed. This was performed in the following way:1.Preservation of structures that are highly similar to the colchicine core. The Tanimoto similarity [43,56,57] threshold value was set to the lowest similarity found among the starting structures to the colchicine core, namely, 0.257 (Appendix A).2.The SYBA selection process, as documented in the study by Vorsilak et al. [42], this stage serves the purpose of eliminating structures that could pose challenges during the synthesis process. The SYBA algorithm yields a numerical SYBA score, wherein higher values indicate greater feasibility for molecular synthesis. The algorithm computes SYBA scores for the initial set of structures, and the lowest recorded score, set at 19.48 (documented in Appendix A), is subsequently employed as the threshold for evaluating newly generated structures. The results of this analysis are stored in Appendix A.3.Stereochemistry [9] selection was performed as the third step; it assumes compounds have the `S’ absolute configuration on the seventh carbon [58]. The `R’ absolute configuration structures do not tend to be biologically active [22,23,24,25,26], thus they are removed from further consideration. The results of this step are stored in Appendix A.4.The process of RI and SI selection is conducted subsequent to the prediction of $IC_{50}$ values, as described in Section 3.3. This pivotal step enables the identification and retention of AI-generated colchicine-based structures that satisfy the prerequisites concerning drug resistance (RI) and specificity towards cancer cell lines (SI). The resultant indices are stored within Appendix A.

Based on t-distributed stochastic neighbor embedding (t-SNE) analysis [64], a dimensionality reduction algorithm, the chemical space of the created structures was compared to the initial structures. This approach let us separate data that could not be divided by a straight line, hence the name “nonlinear dimension reduction“. It gave us the opportunity to comprehend high-dimensional information and transfer it into a low-dimensional space. It reduced the size of each molecule’s molecular fingerprint and presented further similarities between the new structures and the starting structures (Appendix A).

The generated structures were searched for in the PubChem database with the application of the PubChemPy pythonic library [65]. This gave information about whether the structure generated could be found in the PubChem database (Appendix A).

### 3.5. Molecular Docking

The molecular docking studies were conducted via the AutoDock Vina [38] solution. The target protein domain was 1SA0 [41] (PDB [66] ID). The natively present structure and the colchicine itself were docked to the same protein domain, so the result of the calculation conducted could be used as a reference. The investigated active site of the protein domain was the same for each of the structures. The search parameters are given here: center [x, y, z] = [119.743, 92.779, 10.765], size [x, y, z] = [44, 44, 60].

The OpenBabel tool [44,45] was used to create *.pdbqt files, which are necessary for AutoDock Vina, from 3D structures that were previously created with the application of RDKit [9] functionalities. In this manner, it can be performed automatically for many ligands, rather than manually. The protein domain was prepared with the AutoDock Tools 1.5.7 [67].

The following procedure was conducted for the molecular docking:1.Raw 1SA0.pdb structure was downloaded from the PDB;2.Native ligand present inside the pocket was saved separately;3.3D structure of raw colchicine was prepared (Appendix A);4.All the selected new colchicine-based structures were transformed into 3D objects and prepared for molecular docking procedures (Appendix A);5.Molecular docking was conducted (Appendix A), and the results are saved in Appendix A. The visualization of the results is in Appendix A. The final results are stored in Appendix A.

The 2D graphs of the interactions between selected colchicine-based structures have been depicted with the ProteinsPlus web application [68,69,70,71,72,73,74,75,76], which lets users create 2D maps of protein–ligand interactions efficiently.

## 4. Conclusions

The proposed approach opens the opportunity to create a library of new colchicine-based compounds with assigned biological activity for each of the investigated cell lines: A549, BALB/3T3, LoVo/DX, LoVo, and MCF-7. The library created can be in vitro investigated for testing our predictive model capabilities. The methodology presented shows that we can create a large library of structures and conduct multi-step selection. We can use different discriminators, such as similarity of compounds, difficulty of synthesis classification, or chirality of compounds created.

These findings suggest a significant potential for the deliberate selection of chemical structures that align with specific criteria. The RI and SI indices, through their computational calculations, can serve as supplementary criteria for the meticulous curation of AI-generated colchicine-based compounds, facilitating their subsequent synthesis and experimental validation.

Our methodology shown here can be used in other quantitative structure–activity relationship (QSAR) studies. In this study, we evaluated various ML approaches: the RF, DT, MLR, KNN, and SVM ML models. Therefore, we could select the best solution for predicting the half-maximal inhibitory concentration value ($IC_{50}$), for five cell lines of the compounds proposed. The created RF models performed quite well with training and testing data, although the distribution was not pure Gaussian. Surprisingly, it was found that some capabilities for the recognition of $IC_{50}$ patterns were gathered by the RF models for each of the cell lines analyzed.

This model was designed so that it works with colchicine-based compounds, so it has not been used with other structures. The machine learning (ML) models have much higher certainty in the results for more similar structures. If one wanted to predict $IC_{50}$ values for totally different compounds, these would be less certain. This means that we can be more sure of the models’ predictions if the structure we are considering is closer, in chemical space, to the training data. This is the limitation of the model.

## Figures and Tables

**Figure 1 pharmaceuticals-17-00173-f001:**
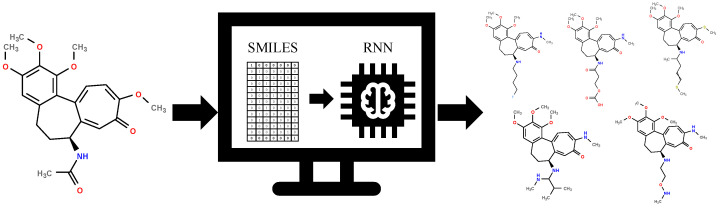
The general scheme of new colchicine-based structures library creation.

**Figure 2 pharmaceuticals-17-00173-f002:**
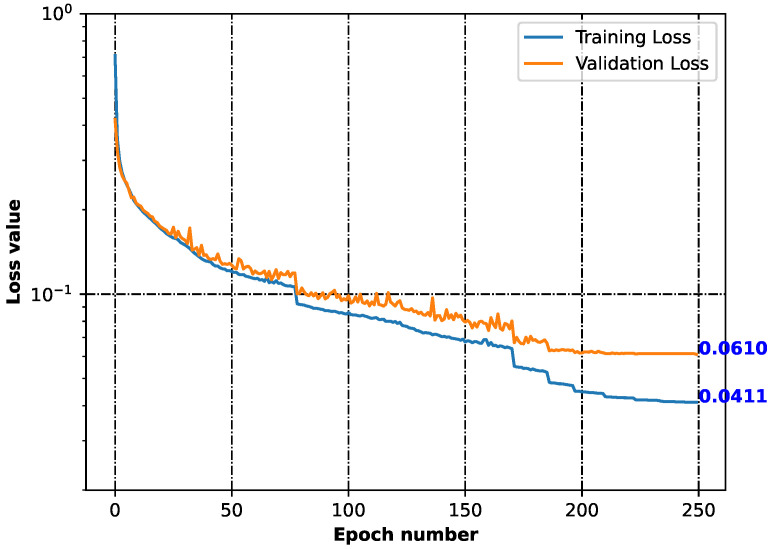
Training and validation losses of the neural network minimization. Both parameters dropping indicates that the model was learning how to generalize from the input.

**Figure 8 pharmaceuticals-17-00173-f008:**
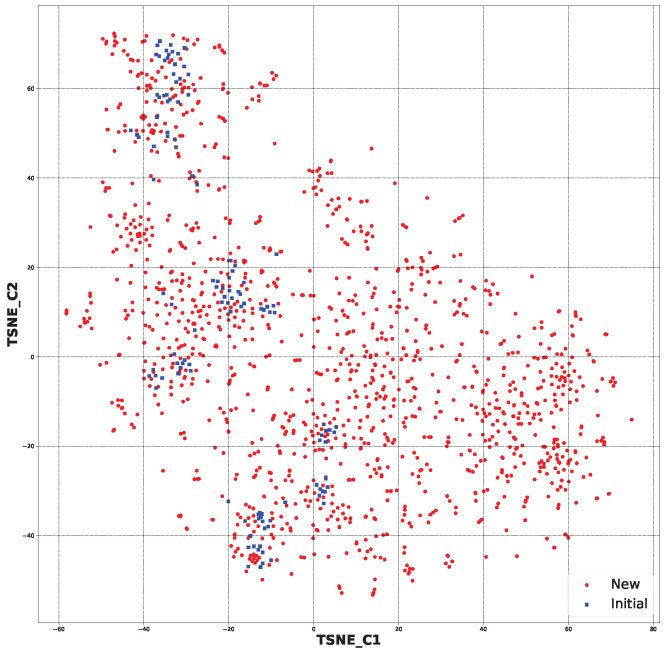
The chemical space for all the newly generated structures (1356) and initial ones (120) based on molecular fingerprints (Appendix A).

**Figure 9 pharmaceuticals-17-00173-f009:**
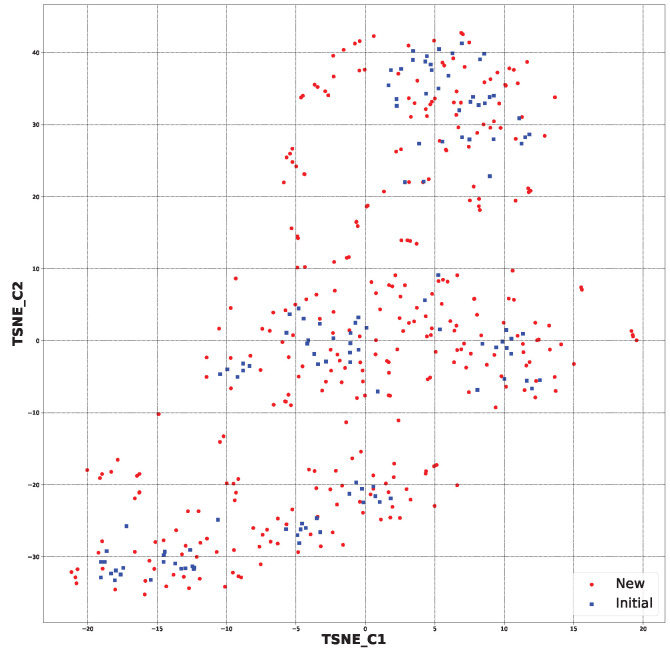
The chemical space for the SYBA-selected newly generated structures (335) and initial ones (120) based on molecular fingerprints (Appendix A).

**Figure 10 pharmaceuticals-17-00173-f010:**
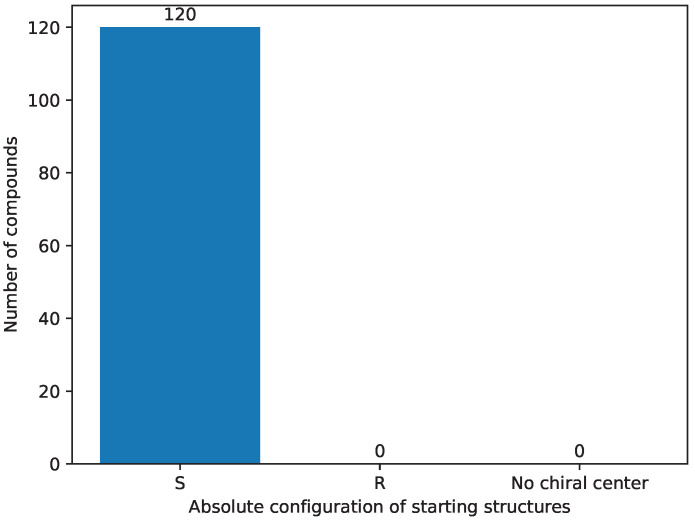
The absolute configuration of starting chemical structures’ distribution.

**Figure 11 pharmaceuticals-17-00173-f011:**
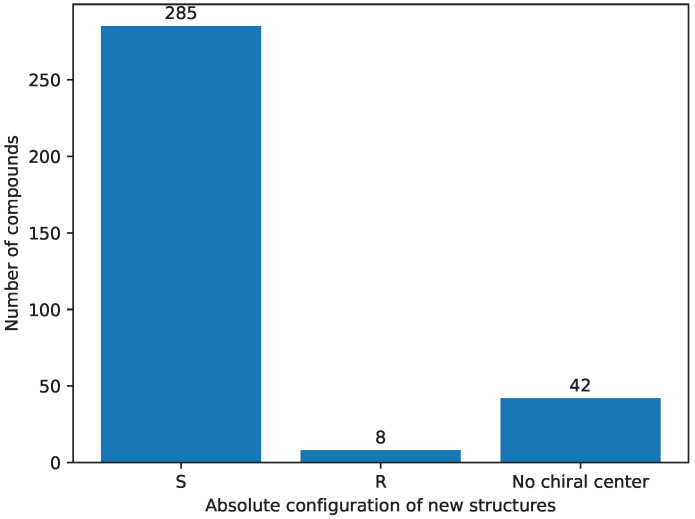
The absolute configuration of newly generated chemical structures’ distribution.

**Figure 12 pharmaceuticals-17-00173-f012:**
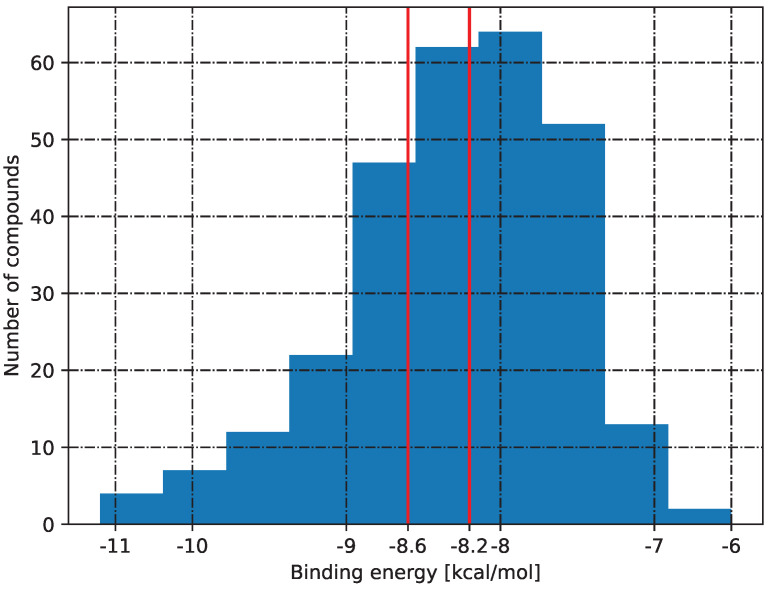
The binding energies of 285 structures that were selected from the previous step (Section 2.4 and Section 3.4). Red lines indicates calculated binding energy of ligand with the protein target.

**Figure 13 pharmaceuticals-17-00173-f013:**
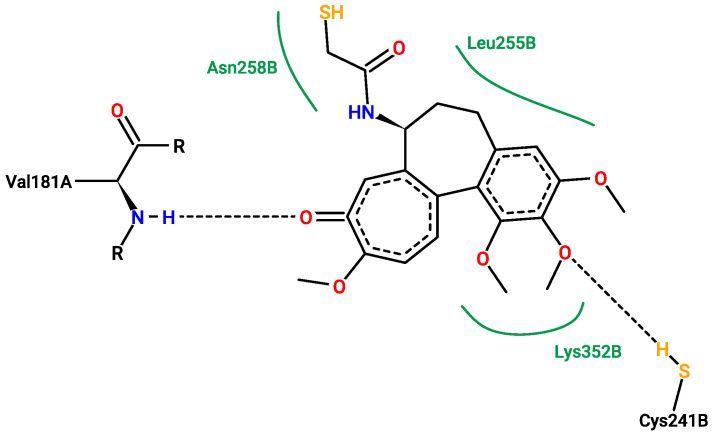
Redocked native structure with binding energy of −8.6 kcal/mol. The picture depitcs possible interactions between the natively present structure in 1SA0 protein’s active site and the active site itself.

**Figure 14 pharmaceuticals-17-00173-f014:**
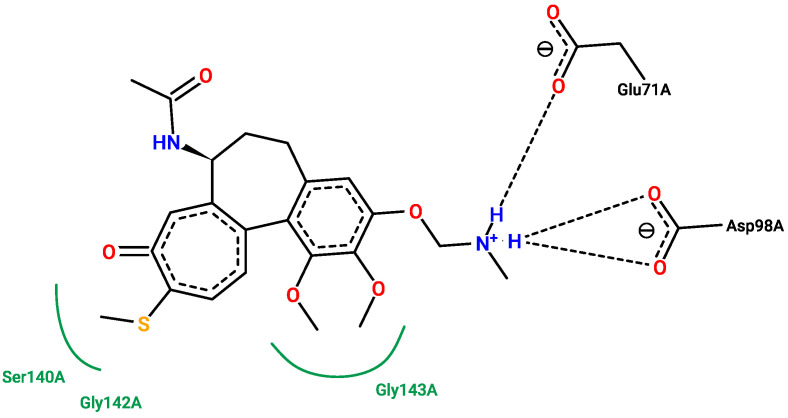
New structure number 113 with binding energy −7.4 kcal/mol. The picture depicts possible interactions between AI-created structure number 113 and the active site of the 1SA0 protein domain.

**Figure 15 pharmaceuticals-17-00173-f015:**
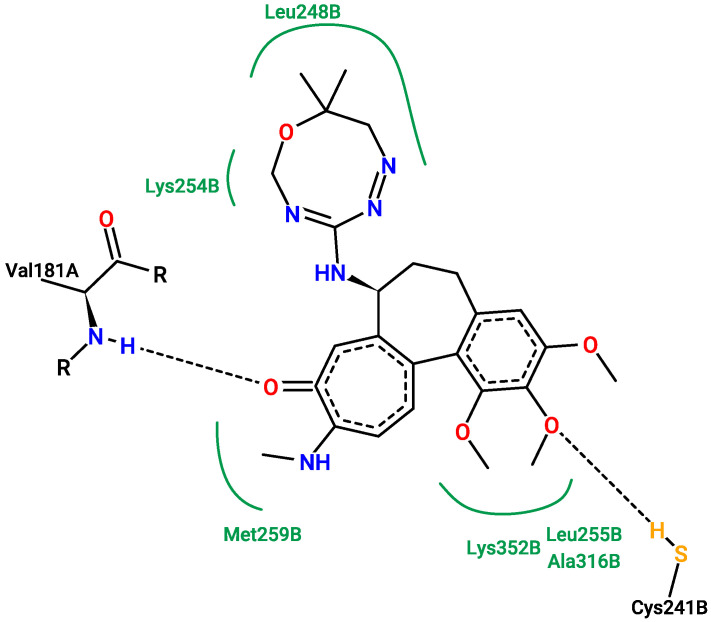
New structure number 73 with binding energy −10.2 kcal/mol. The picture depicts possible interactions between AI-created structure number 73 and the active site of the 1SA0 protein domain.

**Figure 16 pharmaceuticals-17-00173-f016:**
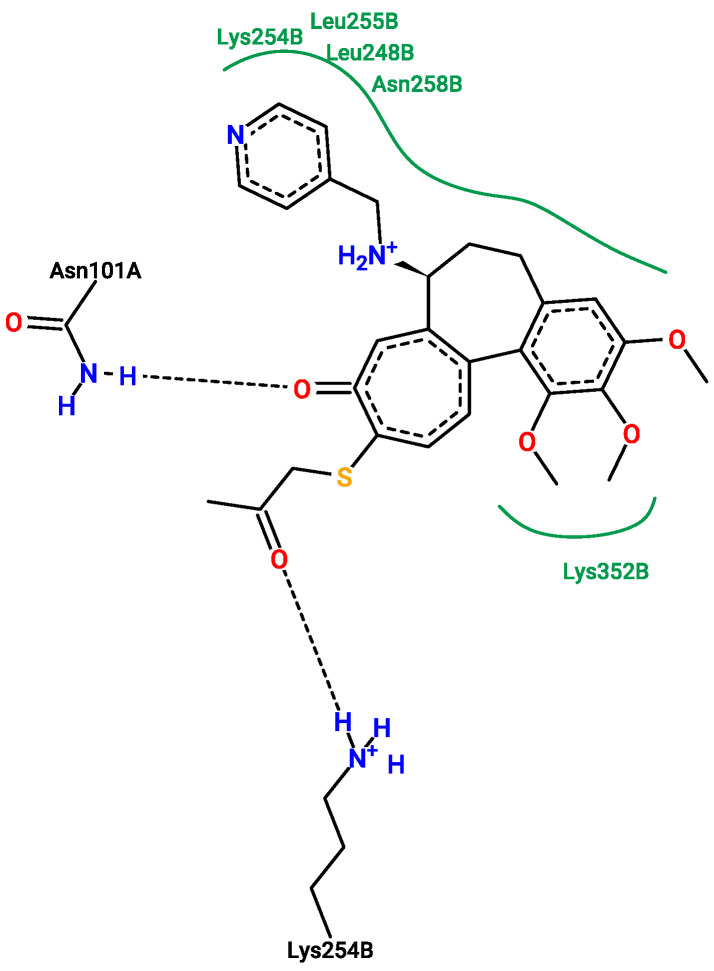
New structure number 162 with binding energy −8.6 kcal/mol. The picture depicts possible interactions between AI-created structure number 162 and the active site of the 1SA0 protein domain.

**Figure 17 pharmaceuticals-17-00173-f017:**
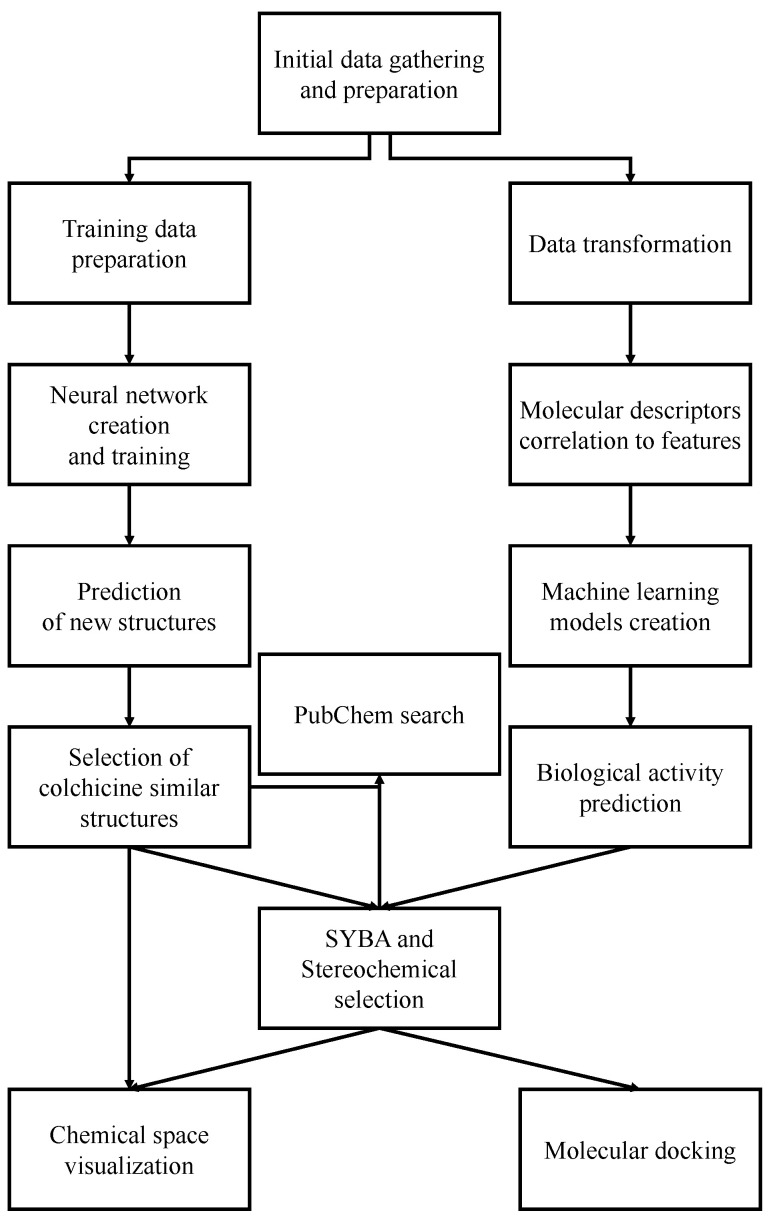
The overall workflow of presented studies.

**Table 1 pharmaceuticals-17-00173-t001:** The structures of selected AI-generated colchicine-based compounds. The table contains two starting structures with the assigned anticancer activities, Tanimoto similarity, and SMILES codes. For the newly generated structures $IC_{50}$ values are predicted and not tested yet.

Starting structures ^1^	
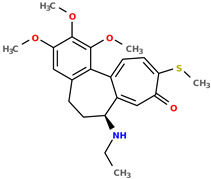 $IC_{50}$ (nM): A549: 10.8 ± 1.4, MCF-7: 10.3 ± 0.4, LoVo: 6.5 ± 1.9, LoVo/DX: 54.9 ± 22.0, BALB/3T3: 10.2 ± 1.9 SMILES: COc2c3C1=CC=C(SC)C(=O)C=C1[C@H](CCc3cc(OC)c2OC)NCC	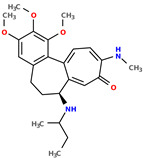 $IC_{50}$ (nM): A549: 9.6 ± 1.3, MCF-7: 9.7 ± 1.5, LoVo: 7.8 ± 1.0, LoVo/DX: 8.5 ± 1.1, BALB/3T3: 7.5 ± 1.5 SMILES: CC(CC)N[C@H]2CCc3cc(OC)c(OC)c(OC)c3C1=CC=C(NC)C(=O)C=C12
Newly proposed structures ^1^	
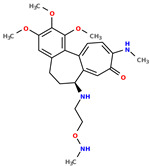 Tanimoto similarity: 0.844 $IC_{50}$ (nM): A549: 8.3, MCF-7: 8.4, LoVo: 6.2, LoVo/DX: 94.3, BALB/3T3: 10.9 SMILES: COC1=C2C3=CC=C(NC)C(=O)C=C3[C@H1](CCC2=CC(OC)=C1OC)NCCONC	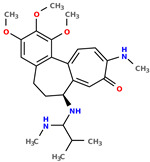 Tanimoto similarity: 0.931 $IC_{50}$ (nM): A549: 11.3, MCF-7: 7.9, LoVo: 9.6, LoVo/DX: 129.6, BALB/3T3: 21.3 SMILES: CNC(C(C)C)N[C@@H1]1C=2C(C3=C(C=C(OC)C(OC)=C3OC)CC1)=CC=C(C(=O)C=2)NC
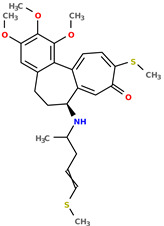 Tanimoto similarity: 0.953 $IC_{50}$ (nM): A549: 11.2, MCF-7: 12.7, LoVo: 14.5, LoVo/DX: 71.1, BALB/3T3: 10.2 SMILES: C1=2[C@H1](CCC3=CC(OC)=C(OC)C(OC)=C3C1=CC=C(SC)C(=O)C=2)NC(C)CC=CSC	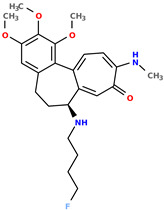 Tanimoto similarity: 0.978 $IC_{50}$ (nM): A549: 15.9, MCF-7: 8.6, LoVo: 9.6, LoVo/DX: 21.6, BALB/3T3: 23.5 SMILES: N([C@H1]1CCC2=CC(OC)=C(OC)C(OC)=C2C3=CC=C(NC)C(C=C31)=O)CCCCF
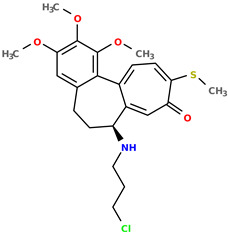 Tanimoto similarity: 0.983 $IC_{50}$ (nM): A549: 10.6, MCF-7: 12.2, LoVo: 14.5, LoVo/DX: 18.1, BALB/3T3: 11.5 SMILES: COC1=C2C3=CC=C(SC)C(=O)C=C3[C@H1](CCC2=CC(OC)=C1OC)NCCCCl	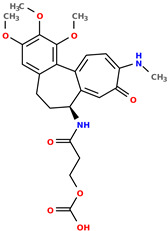 Tanimoto similarity: 0.931 $IC_{50}$ (nM): A549: 900, MCF-7: 4259, LoVo: 345, LoVo/DX: 9490, BALB/3T3: 835 SMILES: O=C(N[C@H1]1CCC2=CC(OC)=C(OC)C(OC)=C2C3=CC=C(NC)C(=O)C=C31)CCOC(=O)O

^1^ Graphical representations of chemical structures were prepared with Open Babel software version 3.1.1 [44,45].

**Table 2 pharmaceuticals-17-00173-t002:** Predictive model approaches investigated for the prediction of given cell lines’ biological activities (Appendix A).

Target	Methodology	Random State	Number of Features	Correlation Threshold	Overall R Score	MSE	MAE	RMSE
A549	RF	15	5	0.51	0.944	0.099	0.205	0.314
BALB/3T3	RF	15	6	0.51	0.950	0.075	0.183	0.274
LoVo/DX	RF	42	5	0.63	0.950	0.089	0.217	0.299
LoVo	RF	28	2	0.54	0.865	0.206	0.320	0.453
MCF-7	RF	15	4	0.51	0.883	0.200	0.258	0.447

## Data Availability

All publication-related information can be accessed at this address: https://github.com/XDamianX-coder/Colchicine_ML (accessed on 18 January 2023). The Appendix A can also be downloaded using the link provided above. Molecular docking results will be shared on request.

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
