# Peer review of "Machine Learning Application for Medicinal Chemistry: Colchicine Case, New Structures, and Anticancer Activity Prediction"

_pharmaceuticals, 2024, doi:10.3390/ph17020173_

Round 1

Reviewer 1 Report

Comments and Suggestions for Authors

Damian Nowak et al. are reporting their application of ML in medicinal chemistry. Overall, this manuscript is well-prepared. The following are specific comments and questions to share.

1.         A recurrent neural network was used in this study to realize the SMILES-based de novo generation of novel molecules. Before diving into generated molecules (Table 1) and the model’s training loss (Figure 1), it will be beneficial to discuss and describe the model’s architecture and performance. A potential graphical illustration of the architecture will provide readers an intuitional idea about the model. Metrics such as validity, novelty, uniqueness, synthetic feasibility, etc. will provide a comprehensive overall picture about the model’s performance. Here is an example RNN(LSTM)-based generative modeling study that may function as a reference https://doi.org/10.3390/cells11050915.

2.         In Table 1, should the SEM or STDEV of IC50 values to be reported as well?

3.         Molecular docking studies were conducted to structurally predict and assess receptor-ligand interactions. Figures 12-15 provided clear and informative contact information. Would it be possible to also generate and include the 3D views of ligands inside the binding pocket? There are chiral centers and rotatable bonds. 3D views will help the reviewer to better appreciate the conformational changes involved in the binding process. 

4.         MD simulations can potentially be continued as a bonus to evaluate the receptor-ligand interactions in a dynamical perspective to validate whether observed key contacts in docking can be stably maintained.  

A major revision would be suggested if the above-mentioned comments and questions can be addressed appropriately.

Comments on the Quality of English Language

Minor editing of English language required

Reviewer 2 Report

Comments and Suggestions for Authors

Manuscript pharmaceuticals-2756242 contains interesting results, but there are shortcomings in evaluation and model selection and in established parts that need to be fixed.

This manuscript does not contain all mentioned information, like the files mentioned in pharmaceuticals-2756242-supplementary.docx.

Without this, the work cannot be properly assessed/evaluated.

The introduction of the manuscript contains sections (especially lines 66-98) that look more like the Methods section than the Introduction section. That needs to be fixed.

Also, the part "The second part is related to the biological 102

activity. Machine Learning (ML) models predictions." must be corrected. The same is true for the sentence starting with " The selection of the 1SA0 protein domain is predicated on the recognition that colchicine..."

The Introduction of the manuscript needs to be completely improved.

Authors should carefully check other parts of the text and correct descriptions where necessary.

In line 258 "All the data can be viewed in File S1." – File S1 and missing in Suppl. inf.

It is similar for all other supplementary files (they aRE MISSING).

The terms "Random state", "Random state 15, 28, 42..." must be explained more precisely.

The main problem is an insufficiently good choice of parameters for evaluating the quality of the model. The authors chose the correlation coefficient (Figures 2-6). For example, in the paper https://doi.org/10.5562/cca3551 (and there are other literature sources) it was shown that the correlation coefficient is not a good measure for assessing the quality of the model in the training phase, but especially in the prediction phase - which is a more important criterion (e.g. pages 384 -386). The use of standard error of estimate or root-mean-squared-error (RMSE) as a better statistical measure is also explained and suggested. It may happen that the conclusions obtained on the basis of the correlation coefficient are similar to those obtained on the basis of RMSE in training/validation/prediction, but it is necessary to calculate, for each model (from Figures 2-6), in addition to the existing correlation coefficients, corresponding RMSE values. A part of those values can be given in the form of tables – in the main text or in supplement.

Figures 2-6 show that the models are probably overfitted on the training set (at least for DT and RF). This can be seen by the high values of R and by the fairly large differences between the corresponding values on the training and test sets - which still seem too large. The authors should comment on the R-values comparisons, but also the RMSEs, which they should (and can very easily) calculate.

Also, many of the other models in Figures 2-6 are not good models - especially those that have very small R values for the test set. That should also be commented on.

Also, for each model in the case of training (on the training set) and prediction on the test set, it is necessary to clearly state which molecules are in each set and to provide the structures and experimental and predicted values for each set of molecules in the structure supplement. Also, it is necessary to provide in the supplement the values of all features used in the models, or at least in the best models. Perhaps the authors provided some of that information, but it was not made available to me on the pages of the manuscript.

Section 2.4. Data selection must be improved.

"The first step of selection has been conducted with Tanimoto similarity [43,51,52]." - it is not clear - about the selection of what is being done, i.e. what data?

Comments on the Quality of English Language

-

Round 2

Reviewer 1 Report

Comments and Suggestions for Authors

Authors have appropriately addressed comments and questions the reviewer posted.

Comments on the Quality of English Language

Only minor editing should be fine. 

Reviewer 2 Report

Comments and Suggestions for Authors

Authors need to improve the manuscript to make it more readable.

There are too many additional files, and many of them can be merged into one Excel file, or into an Excel file with multiple sheets.

In fact, the data is too segmented, and all that can be merged into one Sheet Excel file should be merged.

This is the practice in journals, MDPI also has this practice in its journals and Pharmaceuticals, and authors should respect it - otherwise, the manuscript and additional data are very difficult to follow. Namely, in "Files - colchicine AI.docx" there are links to 70 files - which is a huge number.

Therefore, the authors should look in the published articles in Pharmaceuticals for acceptable practices in the creation of supplementary materials.

All data that can be given in Supplementary materials (eg in Excel file(s), or other formats) as authors normally do in published papers, should be given that way. This ensures that this data is available for many years. Namely, now the authors provide data that can be included in the Supplementary materials and linked to the article on the MDPI website - on his account at the GITHUB. However, if that account will be unavailable in 5, 10 or 20 years, this data would be unavailable to readers, and the article would be difficult (or completely impossible) to follow.

Additionally, this way of displaying Supplementary materials takes too much time for the reviewer, because just imagine how much time it takes to open 70 files via the links provided by the authors in Files - colchicine AI.docx. (?)

Also, in suppl. files some terms are not in English (but Polish), such as Atywność in Kolchicyna_prepared_data.xlsx.

The authors should have marked in the manuscript with colour the newly added parts of the manuscript or those in which they made significant changes. This would make it easier to follow the revision of the manuscript.

Regarding "Ad 6." it seems in the Figures that the relationship shown via RMSE and the comparison of the results on the train and on the test set is better than the one shown via R.

An example where this is best seen is Figure 7, for the Rftrain and Rftest model. When displaying via R, for random state 42, the unreliability of the display via R can be seen. It is similar in Figure 6 for Random state 15 for RF models, as well as Figure 5 for Random state 28,

  Authors can comment on this in the manuscript as well as cite the selected reference(s) in which the advantage of RMSE compared to R (correlation coefficient) is explained.

Comments on the Quality of English Language

-
